# ELCC: THE EMERGENT LANGUAGE CORPUS COLLECTION

## ABSTRACT

We introduce the Emergent Language Corpus Collection (ELCC): a collection of corpora generated from open source implementations of emergent communication systems across the literature. These systems include a variety of signalling game environments as well as more complex tasks environments like a social deduction game and embodied navigation. Each corpus is annotated with metadata describing the characteristics of the source system as well as a suite of analyses of the corpus (e.g., size, entropy, average message length, performance as transfer learning data). Currently, research studying emergent languages requires directly running different systems which takes time away from actual analyses of such languages, makes studies which compare diverse emergent languages rare, and presents a barrier to entry for researchers without a background in deep learning. The availability of a substantial collection of well-documented emergent language corpora, then, will enable research which can analyze a wider variety of emergent languages; this more effectively uncovers general principles in emergent communication rather than artifacts of particular environments. We provide some quantitative and qualitative analyses with ELCC to demonstrate potential use cases of the resource in this vein.

## 1 INTRODUCTION

When Boldt and Mortensen (2024a) introduced the metric called XferBench, they raised a question that they apparently could not answer: how do emergent languages—communication systems that emerge from scratch in agent-based simulations—differ in their "humanlikeness" (as measured by their utility as pretraining data for NLP tasks). It seems likely that they were unable to answer this question because no representative collection of samples from emergent languages existed. The same problem plagues other research programs that seek to make generalizations about emergent languages, as a whole, rather than using a single type of environment as a proof of concept. These include the degree to which emergent languages display entropic patterns similar to those that characterize words in human languages (Ueda et al., 2023) and the kind of syntax that can be detected in emergent languages through grammar induction (van der Wal et al., 2020). We present an initial solution to this problem, namely the Emergent Language Corpus Collection (ELCC), a collection of 73 corpora generated from 7 representative emergent communication systems (ECSs).[1] Prior to this work, comparing emergent languages entailed extensive work getting free and open source simulations to run—managing dependencies, manipulating output formats, etc.—before any data could even be generated. The current work allows investigators, even those with very limited software engineering knowledge, to analyze a wide range of emergent languages straightforwardly, plowing over a barrier that has held back comparative emergent language research from its inception. ELCC will be published with data and code licensed under the CC BY 4.0 and MIT licenses, respectively (data and code included in supplementary material).

We discuss related work in Section 2. Section 3 lays out the design of ELCC while Section 4 describes the content of the collection. Section 5 demonstrates some of the types of analyses enabled by ELCC. Section 6 presents some brief analyses, discussion, and future work related to ELCC. Finally, we conclude in Section 7.

---

[1]Emergent communications systems are more commonly referred to as simply "environments"; we choose to use the term "system" in order to emphasize that what goes into producing an emergent language is more than just an environment including also the architecture of the agents, optimization procedure, datasets, and more.

```
data/ ..................................................................... top-level directory
  ecs-1/ ....................................................... directory for a particular ECS
    metadta.yml ................................................. metadata about the ECS
    code/ ....................................... directory containing files to produce the data
    data/ ........................................ directory containing corpus and metadata files
      hparams-1/ ............................. directory for run with specific hyperparameters
        corpus.jsonl ........................................................... corpus data
        metadata.json ........................... metadata specific for corpus (e.g., metrics)
      hparams-2/ ................................................................... as above
      hparams-n/ ................................................................... as above
  ecs-2/ ......................................................................... as above
  ecs-n/ ......................................................................... as above
```

Figure 1: The file structure of ELCC.

**Contributions**   The primary contribution of this paper is as a first-of-its kind data resource which will enable broader engagement and new research directions within the field of emergent communication. Additionally, code published for reproducing the data resource also improve the reproducibility of existing ECS implementations in the literature, supporting further research beyond just the data resource itself. Finally, the paper demonstrates some of the analyses uniquely made possible by a resource such as ELCC.

## 2   RELATED WORK

**Emergent communication**   There is no direct precedent for this work in the emergent communication literature that we are aware of. Perkins (2021b) introduces the TexRel dataset, but this is a dataset of observations for training ECSs, not data generated by them. Some papers do provide the emergent language corpora generated from their experiments (e.g., Yao et al. (2022a)), although these papers are few in number and only include the particular ECS used in the paper. At a high level, the EGG framework (Kharitonov et al., 2021) strives to make emergent languages easily accessible, though instead of providing corpora directly, it provides a framework for implementing ECSs. Thus, while EGG is useful for someone building new systems entirely, it is not as geared towards research projects aiming directly at analyzing emergent languages themselves.

**Data resources**   At a high level, ELCC is a collection of datasets, each of which represent a particular instance of a phenomenon (emergent communication, in this case). On a this structural level, ELCC is analogous to a collection of different human languages in a multi-lingual dataset. ELCC, though, focuses more on a particular phenomenon of scientific interest, and, in this way, would be more analogous to work such as Blum et al. (2023), which presents a collection of grammar snapshot pairs for 52 different languages as instances of diachronic language change. Similarly, Zheng et al. (2024) present a dataset of conversations from Chatbot Arena, where "text generated by different LLMs" is the phenomenon of interest. Furthermore, insofar as ELCC documents the basic typology of different ECSs, it is similar to the World Atlas of Language Structures (WALS) (Dryer and Haspelmath, 2013).

## 3   DESIGN

### 3.1   FORMAT

ELCC is a collection of ECSs, each of which has one or more associated *variants* which correspond to runs of the system with different hyperparameter settings (e.g., different random seed, message length, dataset). Each variant has metadata along with the corpus generated from its settings. Each ECS has its own metadata as well and code to generate the corpus and metadata of each variant. The file structure of ELCC is illustrated in Figure 1.

**ECS metadata**   Environment metadata provides a basic snapshot of a given system and where it falls in the taxonomy of ECSs. As the collection grows, this structure makes it easier to ascertain the contents of the collection and easily find the most relevant corpora for a given purpose. This metadata will also serve as the foundation for future analyses of the corpora by looking at how the characteristics of an ECS relate to the properties of its output. These metadata include:

- Source information including the original repository and paper of the ECS.
- High-level taxonomic information like game type and subtype.
- Characteristics of observation; including natural versus synthetic data, continuous versus discrete observations.
- Characteristics of the agents; including population size, presence of multiple utterances per episode, presence of agents that send *and* receive messages.
- Free-form information specifying the particular variants of the ECS and general notes about the ELCC entry.

A complete description is given in Appendix A. These metadata are stored as YAML files in each ECS directory. A Python script is provided to validate these entries against a schema. See Appendix B for an example of such a metadata file.

**Corpus**   Each *corpus* comprises a list of *lines* each of which is, itself, an array of *tokens* represented as integers. Each line corresponds to a single episode or round in the particular ECS. In the case of multi-step or multi-agent systems, this might comprise multiple individual utterances which are then concatenated together to form the line (no separation tokens are added). Each corpus is generated from a single run of the ECS; that is, they are never aggregated from distinct runs of the ECS.

Concretely, a *corpus* is formatted as a JSON lines (JSONL) file where each *line* is a JSON array of integer *tokens* (see Figure 3 for an example of the format). There are a few advantages of JSONL: (1) it is a human-readable format, (2) it is JSON-based, meaning it is standardized and has wide support across programming languages, and (3) it is line-based, meaning it is easy to process with command line tools.[2] Corpora are also available as single JSON objects (i.e., and array of arrays), accessible via the Croissant ecosystem (Akhtar et al., 2024).

**Corpus analysis**   For each corpus in ELCC we run a suite of analyses to produce a quantitative snapshot. This suite metrics is intended not only to paint a robust a picture of the corpus but also to serve as jumping-off point for future analyses on the corpora. Specifically, we apply the following to each corpus: token count, unique tokens, line count, unique lines, tokens per line, tokens per line stand deviation, 1-gram entropy, normalized 1-gram entropy, entropy per line, 2-gram entropy, 2-gram conditional entropy, EoS token present, and EoS padding. *Normalized* 1-*gram entropy* is computed as 1-*gram entropy* divided by the maximum entropy given the number of unique tokens in that corpus.

We consider an EoS (end-of-sentence) token to be present when: (1) every line ends with token consistent across the entire corpora, and (2) the first occurrence of this token in a line is only ever followed by more of the same token. For example, `0` could be an EoS token in the corpus `[[1,2,0],[1,0,0]]` but not `[[1,2,0],[0,1,0]]`. EoS padding is defined as a corpus having an EoS token, all lines being the same length, and the EoS token occurs more than once in a line at least once in the corpus.

Additionally, each corpus also has a small amount of metadata copied directly from the output of the ECS; for example, this might include the success rate in a signalling game environment. We do not standardize this because it can vary widely from ECS to ECS, though it can still be useful for comparison to other results among variants within an ECS.

**Reproducibility**   ELCC is designed with reproducibility in mind. With each ECS, code is included to reproduce the corpora and analysis metadata. Not only does this make ELCC reproducible, but it sometimes helps the reproducibility of the underlying implementation insofar as it fixes bugs, specifies Python environments, and provides examples of how to run an experiment with a certain set of hyperparameters. Nevertheless, in this code, we have tried to keep as close to the original implementations as possible. When the underlying implementation supports it, we set the random seed (or keep the default) for the sake of consistency, although many systems do not offer ways to easily set this.

---

[2]E.g., Creating a 100-line random sample of a dataset could be done with `shuf dataset.jsonl | head -n 100 > sample.jsonl`

| Source | Type | Data source | Multi-agent | Multi-step | $n$ corp. |
|--------|------|-------------|-------------|------------|-----------|
| Kharitonov et al. (2021) | signalling | synthetic | No | No | 15 |
| Yao et al. (2022a) | signalling | natural | No | No | 2 |
| Mu and Goodman (2021b) | signalling | both | No | No | 6 |
| Chaabouni et al. (2022) | signalling | natural | Yes | No | 5 |
| Unger and Bruni (2020) | navigation | synthetic | No | Yes | 18 |
| Boldt and Mortensen (2022) | navigation | syntehtic | No | Yes | 20 |
| Brandizzi et al. (2022) | conversation | — | Yes | Yes | 7 |

Table 1: Taxonomic summary the contents of ELCC.

## 4 CONTENT

ELCC contains 73 corpora across 8 ECSs taken from the literature for which free and open source implementations were available. With our selection we sought to capture variation across a three distinct dimensions:

1. Variation across ECSs generally, including elements like game types, message structure, data sources, and implementation details.

2. Variation among different hyperparameter settings within an ECS, including message length, vocabulary size, dataset, and game difficulty.

3. Variation within a particular hyperparameter setting that comes from inherent stochasticity in the system; this is useful for gauging the stability or convergence of an ECS.

Table 1 shows an overview of the taxonomy of ELCC based on the ECS-level metadata. In addition to this, Table 2 provides a quantitative summary of the corpus-level metrics described in Section 3.1. We separate the discussion of particular systems into two subsections: signalling games (Section 4.2) and its variations which represent a large proportion of system discussed in the literature and other games (Section 4.3) which go beyond the standard signalling framework.

### 4.1 SCOPE

The scope of the contents of ELCC is largely the same as discussed in reviews such as Lazaridou and Baroni (2020) and Boldt and Mortensen (2024b, Section 1.2). This comprises agent-based models for simulating the formation of "natural" language from scratch using deep neural networks. Importantly, *from scratch* means that the models are not pretrained or tuned on human language. Typically, such simulations make use of reinforcement learning to train the neural networks, though this is not a requirement in principle.

One criterion that we do use to filter ECSs for inclusion is its suitability for generating corpora as described above. This requires that the communication channel is discrete, analogous to the distinct words/morphemes which for the units of human language. This excludes a small number of emergent communication papers have approached emergent communication through constrained continuous channels like sketching (Mihai and Hare, 2021b) or acoustic-like signals (Eloff et al., 2023). Other systems use discrete communication but are in essence one token per episode (e.g., Tucker et al. (2021b)) which would not form a suitable corpus for addressing most research questions.

### 4.2 SIGNALLING GAMES

The *signalling game* (or *reference game*) (Lewis, 1969) represents a plurality, if not majority, of the systems present in the literature. A brief, non-exhaustive review of the literature yielded 43 papers which use minor variations of the signalling game, a large number considering the modest body of emergent communication literature (see Appendix C). The basic format of the signalling game is a single round of *sender* agent making an observation, passing a message to the *receiver* agent, and the receiver performing an action based on the information from the message. The popularity of this game is, in large part, because of its simplicity in both concept and implementation. Experimental variables can be manipulated easily while introducing minimal confounding factors. Furthermore, the

implementations can entirely avoid the difficulties of reinforcement learning by treating the sender and receiver agents as a single neural network, resulting in autoencoder with a discrete bottleneck which can be trained with backpropagation and supervised learning.

The two major subtypes of the signalling game are the *discrimination game* and the *reconstruction game*. In the discrimination game, the receiver must answer a multiple-choice question, that is, select the correct observation from among incorrect "distractors". In the reconstruction game, the receiver must recreate the input directly, similar to the decoder of an autoencoder.

**Vanilla**   For the most basic form of the signalling game, which we term "vanilla", we use the implementation provided in the Emergence of lanGuage in Games (EGG) framework (Kharitonov et al., 2021, MIT license). It is vanilla insofar as it comprises the signalling game with the simplest possible observations (synthetic, concatenated one-hot vectors), a standard agent architecture (i.e., RNNs), and no additional dynamics or variations on the game. Both the discrimination game and the reconstruction game are included. This system provides a good point of comparison for other ECSs which introduce variations on the signalling game. The simplicity of the system additionally makes it easier to vary hyperparameters: for example, the size of the dataset can be scaled arbitrarily and there is no reliance on pretrained embedding models.

**Natural images**   "Linking emergent and natural languages via corpus transfer" (Yao et al., 2022a, MIT license) presents a variant of the signalling game which uses embeddings of natural images as the observations. In particular, the system uses embedded images from the MS-COCO and Conceptual Captions datasets consisting of pictures of everyday scenes. Compared to the uniformly sampled one-hot vectors in the vanilla setting, natural image embeddings are real-valued with a generally smooth probability distribution rather than being binary or categorical. Furthermore, natural data distributions are not uniform and instead have concentrations of probability mass on particular elements; this non-uniform distribution is associated with various features of human language (e.g., human languages' bias towards describing warm colors (Gibson et al., 2017; Zaslavsky et al., 2019)).

**Concept-based observations**   "Emergent communication of generalizations" (Mu and Goodman, 2021b, MIT license) presents a variant of the discrimination signalling game which they term the *concept game*. The concept game changes the way that the sender's observation corresponds with the receiver's observations. In the vanilla discrimination game, the observation the sender sees is exactly the same as the correct observation that the receiver sees. In the concept game, the sender instead observes a set of inputs which share a particular concept (e.g., red triangle and red circle are both red), and the correct observation (among distractors) shown to the receiver contains the same concept (i.e., red) while not being identical to those observed by the sender. The rationale for this system is that the differing observations will encourage the sender to communicate about abstract concepts rather than low-level details about the observation. This ECS also presents the vanilla discrimination game as well as the *set reference game*, which is similar to the reference game except that the whole object is consistent (e.g., different sizes and locations of a red triangle).

**Multi-agent population**   "Emergent communication at scale" (Chaabouni et al., 2022, Apache 2.0-license) presents a signalling game system with populations of agents instead of the standard fixed pair of sender and receiver. For each round of the game, then, a random sender is paired with a random receiver. This adds a degree of realism to the system, as natural human languages are ways developed within a population and not just between two speakers (cf. idioglossia). More specifically, language developing among a population of agents prevents some degree "overfitting" between sender and receiver; in this context, having a population of agents functions as an ensembling approach to regularization.

## 4.3   OTHER GAMES

Considering that the signalling game is close to the simplest possible game for an ECS, moving beyond the signalling game generally entails an increase in complexity. There is no limit to the theoretical diversity of games, although some of the most common games that we see in the literature are conversation-based games (e.g., negotiation, social deduction) and navigation games. These games often introduce new aspects to agent interactions like: multi-step episodes, multi-agent interactions, non-linguistic actions, and embodiment.

These kinds of systems, as a whole, are somewhat less popular in the literature. On a practical level, more complex systems are more difficult to implement and even harder to get to converge reliably—many higher-level behaviors, such as planning or inferring other agent's knowledge, are difficult problems for reinforcement learning in general, let alone with discrete multi-agent emergent communication. On a methodological level, more complexity in the ECS makes it harder to formally analyze the system as well as eliminate confounding factors in empirical investigation. With so many moving parts, it can be difficult to prove that some observed effect is not just a result of some seemingly innocent hyperparameter choice (e.g., learning rate, samples in the rollout buffer) (Boldt and Mortensen, 2022). Nevertheless, we have reason to believe that these complexities are critical to understanding and learning human language as a whole (Bisk et al., 2020), meaning that the difficulties of more complex systems are worth overcoming as they are part of the process of creating more human-like emergent languages, which are more informative for learning about human language and more suitable for applications in NLP.

**Grid-world navigation**  "Generalizing Emergent Communication" (Unger and Bruni, 2020, BSD-3-clause license) introduces an ECS which takes some of the basic structure of the signalling game and applies it to a navigation-based system derived from the synthetic Minigrid/BabyAI environment (Chevalier-Boisvert et al., 2018; 2023). A sender with a bird's-eye view of the environment sends messages to a receiver with a limited view who has to navigate to a goal location. Beyond navigation, some environments present a locked door for which the receiver must first pick up a key in order to open. What distinguishes this system most from the signalling game is that it is multi-step and embodied such that the utterances within an episodes are dependent on each other. Among other things, this changes the distribution properties of the utterances. For example, if the receiver is in Room A at timestep $T$, it is more likely to be in Room A at timestep $T + 1$; thus if utterances are describing what room the receiver is in, this means that an utterance at $T + 1$ has less uncertainty given the content of an utterance at $T$. Practically speaking, the multiple utterances in a given episode are concatenated together to form a single line in the corpus in order to maintain the dependence of later utterances on previous ones.

**Continuous navigation**  "Mathematically Modeling the Lexicon Entropy of Emergent Language" (Boldt and Mortensen, 2022, GPL-3.0 license) introduces a simple navigation-based ECS which is situated in a continuous environment. A "blind" receiver is randomly initialized in an obstacle-free environment and must navigate toward a goal zone guided by messages from the sender which observes the position of the receiver relative to the goal. The sender sends a single discrete token at each timestep, and a line in the dataset consists of the utterances from each timestep concatenated together. This system shares the time-dependence between utterances of the grid-world navigation system although with no additional complexity of navigating around obstacle, opening doors, etc. On the other hand, the continuous nature of this environment provides built-in stochasticity since there are (theoretically) infinitely many distinct arrangements of the environment that are possible, allowing for more natural variability in the resulting language.

**Social deduction**  "RLupus: Cooperation through the emergent communication in The Werewolf social deduction game" (Brandizzi et al., 2022, GPL-3.0 license) introduces an ECS based on the social deduction game *Werewolf* (a.k.a., *Mafia*) where, through successive rounds of voting and discussion, the "werewolves" try to eliminate the "villagers" before the villagers figure out who the werewolves are. In a given round, the discussion takes the form of all agents broadcasting a message to all other agents after which a vote is taken on whom to eliminate. As there are multiple rounds in a given game, this system introduces multi-step as well as multi-speaker dynamics into the language. Furthermore, the messages also influence distinct actions in the system (i.e., voting). These additional features in the system add the potential for communication strategies that are shaped by a variety of heterogeneous factors rather than simply the distribution of observations (as in the signalling game).

## 5 ANALYSIS

In this section we give present a brief set of analyses that demonstrate some of the possible insights that can be gained from ELCC. Table 2 shows the five-number summary of the corpus-level metrics in ELCC (full results in Appendix D). The corpora come in all shapes and sizes, so to speak, demonstrating a wide range of token counts, vocabulary sizes, entropies, and so on. The variety,

|                            | min   | 25%    | 50%    | 75%      | max       |
|----------------------------|-------|--------|--------|----------|-----------|
| Token Count                | 48616 | 67248  | 110000 | 1061520  | 42977805  |
| Line Count                 | 999   | 5765   | 10000  | 10000    | 2865187   |
| Tokens per Line            | 5.87  | 7.00   | 11.00  | 33.53    | 7212.72   |
| Tokens per Line SD         | 0.00  | 0.00   | 2.31   | 13.81    | 445.84    |
| Unique Tokens              | 2     | 7      | 10     | 20       | 902       |
| Unique Lines               | 18    | 1253   | 2440   | 4911     | 309405    |
| 1-gram Entropy             | 0.36  | 2.12   | 2.80   | 3.37     | 6.60      |
| 1-gram Normalized Entropy  | 0.16  | 0.71   | 0.82   | 0.90     | 1.00      |
| 2-gram Entropy             | 0.42  | 3.16   | 4.11   | 5.88     | 12.88     |
| 2-gram Conditional Entropy | 0.06  | 0.85   | 1.41   | 2.54     | 6.29      |
| Entropy per Line           | 4.38  | 21.23  | 30.80  | 71.85    | 30233.52  |

Table 2: Five-number summary of the analyses across corpora of ELCC. Entropy in bits.

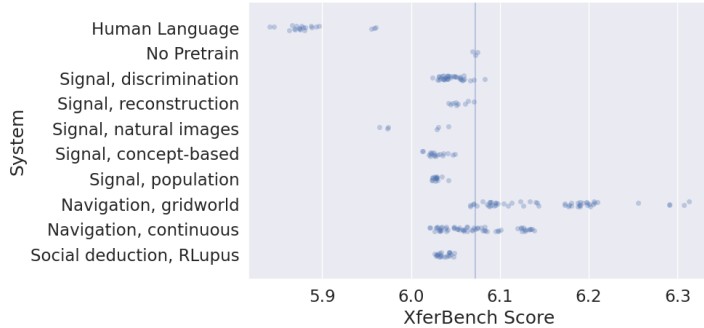

Figure 2: XferBench score across ELCC and human language baselines; lower is better. "No pretrain" baseline illustrated with the line on the plot.

in large part, comes from the diversity of systems included in ELCC rather than variation within a system. Thus research focusing on a single or narrow range of emergent communication systems—the norm prior to ELCC—restricts itself to a limited diversity of corpus "shapes"; ELCC, in turn, provides an easy opportunity to expand the breadth of many such approaches.

The range of analyses ELCC enables is greatly multiplied by a resource like XferBench (Boldt and Mortensen, 2024a), a deep transfer learning-based evaluation metric for emergent languages. This metric quantifies how good a corpus is as pretraining data for a human language-based downstream task, specifically language modelling (thus a lower score is better). XferBench proves to be a particularly powerful for analyzing ELCC because it works in an environment-agnostic way, taking only a corpus of tokenized utterances as input. In fact, ELCC and XferBench permit the first large-scale comparison of emergent language systems with an *evaluative* metric.

**Explaining XferBench's performance**  In addition to the purely descriptive metrics discussed above, we also present evaluative metrics via XferBench in Figure 2. We run XferBench three times for each corpus since there inherent stochasticity in XferBench (results included in supplementary materials). We see that most of the emergent languages occupy a band which slightly outperforms the baselines (i.e., no pretraining at all) while significantly underperforming human languages (exception discussed below). Notably, two of the environments with the worst-performing corpora are the grid-world (Unger and Bruni, 2020) and continuous (Boldt and Mortensen, 2022) navigation environments, while the signalling games perform better consistently.

Inspecting some utterances from the best- and worst- performing corpora, we can see a qualitative difference in Figure 3. The best-performing corpus uses a variety of tokens derived from a large vocabulary (given the high token IDs), while the worst-performing corpus repeats the same two tokens with little variation (this sample is representative of the whole corpus). We hypothesize that

```
[47, 2466, 47, 3923, 3325, 3107,
    3350, 3923, 1216, 3980, 1617,
    3350, 1897, 556, 0]
[3925, 3925, 3925, 3325, 1172,
    2530, 3925, 1209, 3493, 665,
    512, 3923, 2432, 309, 0]
[2128, 2128, 2371, 3925, 946, 512,
    1962, 1288, 2250, 1722, 1722,
    1962, 3755, 2695, 0]
```

(a) Best-performing: signalling game (Yao et al., 2022a) with the COCO dataset.

```
[3, 3, 3, 3, 3, 3, 3, 3, 7, 7, 7,
    7, 7, 7, 7, 7]
[3, 3, 3, 3, 3, 3, 3, 3, 3, 3, 3,
    3, 3, 3, 3, 3]
[3, 3, 3, 3, 3, 3, 3, 3]
```

(b) Worst-performing: BabyAI-based navigation game (Unger and Bruni, 2020) (hyperparameters in text).

Figure 3: Sample utterances from the best and worst performing emergent language corpora on XferBench from ELCC.

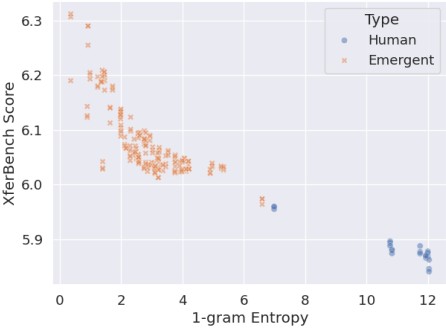

(a) Plot of XferBench score versus unigram entropy for emergent languages and baseline human languages from XferBench.

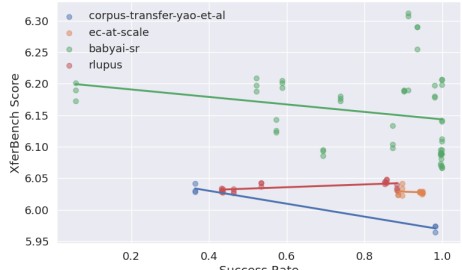

(b) Plot of XferBench score versus success rate, separated by emergent communication system.

Figure 4

pretraining on repetitive strings of a small variety of tokens poorly conditions the model used in XferBench, supported by the fact that the lowest entropy corpora perform the worst on XferBench.

The qualitative analysis suggests that something along the lines of variation or information content might be correlated with XferBench score. To investigate this, we plot two possible explanatory variables against XferBench scores: unigram entropy and task success rate Figure 4. Immediately, we can see that there is a strong correlation between entropy and XferBench score. In fact, this plot gives some insight into the anomalously low score on "Signal, natural images" (Yao et al., 2022a) and anomalously high score for Hindi (an unresolved quandary of the XferBench paper): both of these corpora perform as expected given their entropies. On the other hand, success rate does not seem to be well-correlated with score on XferBench; surprisingly enough, the worst-performing corpus shown above still sported a >90% task success rate!

**Evaluating improvements in ECS design**    Finally, we are also able to use XferBench and ELCC to evaluate some of the innovations in emergent communication system design made by papers contributing to ELCC. Namely, we look at Mu and Goodman (2021b) and "EC at Scale" (Chaabouni et al., 2022). Mu and Goodman (2021b) introduce (as discussed in Section 4.2) a more sophisticated, concept-focused version of the signalling game, comparing it against a vanilla signalling game ("reference") and an intermediate form of the concept version ("set reference"), finding that the introduced games promote more systematic and interpretable emergent languages. On the other hand, Chaabouni et al. (2022) introduces multi-agent populations to the signalling game but does not find that larger populations have a beneficial effect on communication. Looking at the systems' performance XferBench (Figure 5), we can see that the proposed improvements to the signalling game do not have an appreciable effect on XferBench performance in either case. These results do not detract from the original findings; instead, evaluating the design changes with XferBench better contextualizes work, highlighting to what degree certain desirable features of emergent languages (e.g., interpretability, robustness) correspond with suitability for deep transfer learning.

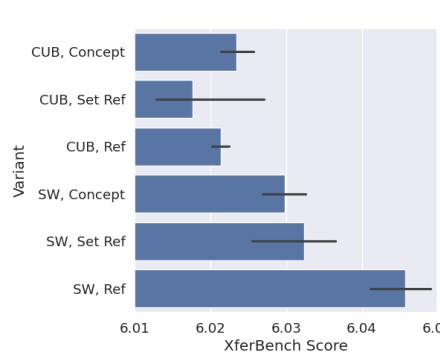

(a) Expected order: concept, set reference, reference (Mu and Goodman, 2021b).

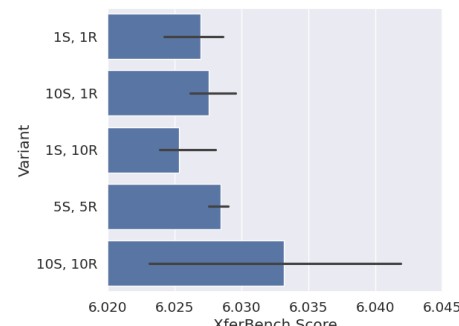

(b) # of senders, # of receivers; more agents expected to perform better than fewer (Chaabouni et al., 2022).

Figure 5: XferBench scores compared to expected order; lower is better.

## 6 DISCUSSION

**Work enabled by ELCC**   In the typical emergent communication paper, only a small amount of time and page count is allocated to analysis with the lion's share being taken up by designing the ECS, implementing it, and running experiments. Even if one reuses an existing implementation, a significant portion of work still goes towards designing and running the experiments, and the analysis is still limited to that single system. While this kind of research is valid and important, it should not be the only paradigm possible within emergent communication research. To this end, ELCC enables research which focus primarily on developing more in-depth analyses across a diverse collection of systems. Furthermore, removing the necessity of implementing and/or running experiments allows researchers without machine learning backgrounds to contribute to emergent communication research from more linguistic angles that otherwise would not be possible.

In particular, ELCC enables work that focuses on the lexical properties of emergent communication, looking at the statical properties and patterns of the surface forms of a given language (e.g., Zipf's law). Ueda et al. (2023) is a prime example of this; this paper investigates whether or not emergent languages obey Harris' Articulation Schema (HAS) by relating conditional entropy to the presence of word boundaries. The paper finds mixed evidence for HAS in emergent languages but only evaluated a handful of settings in a single ECS, yet it could be the case that only systems with certain designs generate languages described by HAS. The variety of systems provided by ELCC could, then, provide more definitive empirical evidence in support or against the presence of HAS in emergent languages. Additionally, ELCC can similarly extend the range of emergent languages evaluated in the context of machine learning, such as Yao et al. (2022a); Boldt and Mortensen (2024a) which look at emergent language's suitability for deep transfer learning to downstream NLP tasks or van der Wal et al. (2020) which analyzes emergent languages with unsupervised grammar induction.

**ECS implementations and reproducibility**   In the process of compiling ELCC, we observed a handful of trends in the implementations of emergent communication systems. A significant proportion of papers do not publish the implementations of experiments, severely limiting the ease of reproducing the results or including such work in a project such as ELCC, considering that a large amount of the work in creating an ECS is not in the design but in the details of implementation. Even when a free and open source implementation is available, many projects suffer from underspecified Python dependencies (i.e., no indication of versions) which can be difficult to reproduce if the project is older than a few years. Furthermore, some projects also fail to specify the particular hyperparameter settings or commands to run the experiments presented in the paper; while these can often be recovered with some investigation, this and the above issue prove to be obstacles which could easily be avoided. For an exemplar of a well-documented, easy-to-run implementation of an ECS and its experiments, see Mu and Goodman (2021b) at `https://github.com/jayelm/emergent-generalization/` which not only provides

dependencies with version and documentation how to download the data but also a complete shell script which executes the commands to reproduce the experiments.

**Future of ELCC**   While ELCC is a complete resource as presented in this paper, ELCC is intended to be an ongoing project which incorporates further ECSs, analyses, and taxonomic features as the body of emergent communication literature and free and open source implementations continues to grow. This approach involves the community not only publishing well-documented implementation of their ECSs but also directly contributing to ELCC in the spirit of scientific collaboration and free and open source software. ELCC, then, is intended to become a hub for a variety of stakeholders in the emergent communication research community, namely a place for: ECS developers to contribute and publicize their work, EC researchers to stay up-to-date on new ECSs, and EC-adjacent researchers to find emergent languages which they can analyze or otherwise use for their own research.

**Limitations**   Emergent communication research is primarily basic research on machine generated data; thus, ELCC has few, if any, direct societal impacts. From a research point of view: while ELCC attempts to provide a representative sample of the ECSs present in the literature, it is not comprehensive collection of all of the open source implementations let alone all ECSs in the literature. This limitation is especially salient in the case of foundational works in EC which have no open source implementations (e.g., Mordatch and Abbeel (2018)). Thus, the contents of ELCC could potentially result in an over-reliance on the particular systems included resulting in an unfamiliarity with the data and limiting research on those currently not included in ELCC. Including the data-generating code and metadata describing the systems in ELCC has partially addressed this issue, and future work adding more open source implementations and reimplementing seminal papers could continue to ameliorate this limitation.

Beyond the variety of systems, in its design ELCC only provides unannotated corpora without any reference to the semantics of the communication, which limits the range of analyses that can be performed (e.g., measures of compositionality are precluded because since it fundamentally a relationship between surface forms and their semantics). In terms of compute resources, we estimate that on the order of 150 GPU-hours (NVIDIA A6000 or equivalents) on an institutional cluster were used in the development of ELCC, and additional 1000 GPU-hours were used to generate the results of XferBench on ELCC. This research could be difficult to reproduce without access to institutional resources.

## 7   CONCLUSION

In this paper, we have introduced ELCC, a collection of emergent language corpora annotated with taxonomic metadata and suite of descriptive metrics derived from free and open source implementations of emergent communication systems introduced in the literature. ELCC also provides code for running these implementations, in turn, making those implementations more reproducible. This collection is the first of its kind in providing easy access to a variety of emergent language corpora. Thus, it enables new kinds of research on emergent communication which involve a wide range of emergent communication which focuses directly on the analysis of the emergent languages themselves.

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

# A    ECS-LEVEL METADATA SPECIFICATION

**source**  The URL for the repository implementing the ECS.

**upstream_source**  The URL of the original repo if **source** is a fork.

**paper**  The URL of the paper documenting the ECS (if any).

**game_type**  The high level category of the game implemented in the ECS; currently one of *signalling*, *conversation*, or *navigation*.

**game_subtype**  A finer-grained categorization of the game, if applicable.

**observation_type**  The type of observation that the agents make; currently either *vector* or *image* (i.e., an image embedding).

**observation_continuous**  Whether or not the observation is continuous as opposed to discrete (e.g., image embeddings versus concatenated one-hot vectors).

**data_source**  Whether the data being communicated about is from a natural source (e.g., pictures), is synthetic, or comes from another source (e.g., in a social deduction game).

**variants**  A dictionary where each entry corresponds to one of the variants of the particular ECS. Each entry in the dictionary contains any relevant hyperparameters that distinguish it from the other variants.

**seeding_available**  Whether or not the ECS implements seeding the random elements of the system.

**multi_step**  Whether or not the ECS has multiple steps per episode.

**symmetric_agents**  Whether or not agents both send and receive messages.

**multi_utterance**  Whether or not multiple utterances are included per line in the dataset.

**more_than_2_agents**  Whether or not the ECS has a population of $>2$ agents.

# B    ECS-LEVEL METADATA EXAMPLE

See Figure 6.

# C    PAPERS BASED ON THE SIGNALLING GAME

Mu and Goodman (2021a); Ohmer et al. (2022); Yao et al. (2022b); Rita et al. (2022a); Ohmer et al. (2021); Łukasz Kuciński et al. (2021); Portelance et al. (2021); Tucker et al. (2021a); Dessì et al. (2021); Bullard et al. (2021); Perkins (2021a); Mihai and Hare (2021a); Denamganaï and Walker (2020); Guo et al. (2020); Li et al. (2020); Rita et al. (2020); Chowdhury et al. (2020a;b); Lan et al. (2020); Chaabouni et al. (2020); Luna et al. (2020); Kharitonov and Baroni (2020); Ren et al. (2020); Słowik et al. (2020); Lowe et al. (2020); Keresztury and Bruni (2020); Dagan et al. (2020); Mihai and Hare (2019); Dessì et al. (2019); Guo et al. (2019); Steinert-Threlkeld (2019); Li and Bowling (2019); Kharitonov et al. (2019); Chaabouni et al. (2019); Khomtchouk and Sudhakaran (2018); Bouchacourt and Baroni (2018); Lazaridou et al. (2018); Havrylov and Titov (2017); Lazaridou et al. (2016); Mahaut et al. (2023); Carmeli et al. (2022); Rita et al. (2022b); Downey et al. (2022)

# D    PER SYSTEM ANALYSIS

See Tables 3 to 6.

```
origin:
  upstream_source:
    https://github.com/google-deepmind/emergent_communication...
  paper: https://openreview.net/forum?id=AUGBfDIV9rL
system:
  game_type: signalling
  data_source: natural
  game_subtype: discrimination
  observation_type: image
  observation_continuous: true
  seeding_available: true
  multi_step: false
  more_than_2_agents: true
  multi_utterance: false
  symmetric_agents: false
  variants:
    imagenet-1x10:
      n_receivers: 10
      n_senders: 1
    imagenet-10x10:
      n_receivers: 10
      n_senders: 10
    imagenet-5x5:
      n_receivers: 5
      n_senders: 5
    imagenet-1x1:
      n_receivers: 1
      n_senders: 1
    imagenet-10x1:
      n_receivers: 1
      n_senders: 10
```

Figure 6: Example of an ECS metadata file in the YAML format.

| name | Token Count | Line Count | Tokens per Line | Tokens per Line SD |
|---|---|---|---|---|
| babyai-sr/GoToObj | 130648 | 6116 | 21.361674 | 12.470737 |
| babyai-sr/GoToObjLocked | 272712 | 5629 | 48.447682 | 15.939260 |
| babyai-sr/GoToObjLocked_ambiguous | 229504 | 5507 | 41.674959 | 17.414703 |
| babyai-sr/GoToObjLocked_ambiguous-freq_1 | 2605112 | 5179 | 503.014482 | 45.179870 |
| babyai-sr/GoToObjLocked_ambiguous-freq_2 | 1061520 | 5396 | 196.723499 | 70.458489 |
| babyai-sr/GoToObjLocked_ambiguous-freq_32 | 67248 | 5496 | 12.235808 | 3.993043 |
| babyai-sr/GoToObjLocked_ambiguous-freq_4 | 402248 | 5728 | 70.224860 | 30.849604 |
| babyai-sr/GoToObjLocked_ambiguous-msg_16 | 511840 | 5514 | 92.825535 | 33.765546 |
| babyai-sr/GoToObjLocked_ambiguous-msg_32 | 855744 | 5508 | 155.363834 | 58.659355 |
| babyai-sr/GoToObjLocked_ambiguous-msg_4 | 103228 | 5730 | 18.015358 | 6.603910 |
| babyai-sr/GoToObjUnlocked | 118752 | 6077 | 19.541221 | 7.060342 |
| babyai-sr/GoToObjUnlocked-freq_1 | 1666456 | 6006 | 277.465201 | 205.399768 |
| babyai-sr/GoToObjUnlocked-freq_2 | 333552 | 5777 | 57.737926 | 28.293252 |
| babyai-sr/GoToObjUnlocked-freq_32 | 48616 | 6001 | 8.101316 | 0.894576 |
| babyai-sr/GoToObjUnlocked-freq_4 | 193176 | 5762 | 33.525859 | 13.813609 |
| babyai-sr/GoToObjUnlocked-msg_16 | 273008 | 6038 | 45.214972 | 18.173718 |
| babyai-sr/GoToObjUnlocked-msg_32 | 469440 | 5765 | 81.429315 | 33.131090 |
| babyai-sr/GoToObjUnlocked-msg_4 | 58588 | 5759 | 10.173294 | 4.351285 |
| corpus-transfer-yao-et-al/cc | 42977805 | 2865187 | 15.000000 | 0.000000 |
| corpus-transfer-yao-et-al/coco_2014 | 1241745 | 82783 | 15.000000 | 0.000000 |
| ec-at-scale/imagenet-10x1 | 2500000 | 250000 | 10.000000 | 0.000000 |
| ec-at-scale/imagenet-10x10 | 2500000 | 250000 | 10.000000 | 0.000000 |
| ec-at-scale/imagenet-1x1 | 2500000 | 250000 | 10.000000 | 0.000000 |
| ec-at-scale/imagenet-1x10 | 2500000 | 250000 | 10.000000 | 0.000000 |
| ec-at-scale/imagenet-5x5 | 2500000 | 250000 | 10.000000 | 0.000000 |
| egg-discrimination/4-attr_4-val_3-dist_0-seed | 110000 | 10000 | 11.000000 | 0.000000 |
| egg-discrimination/4-attr_4-val_3-dist_1-seed | 110000 | 10000 | 11.000000 | 0.000000 |
| egg-discrimination/4-attr_4-val_3-dist_2-seed | 110000 | 10000 | 11.000000 | 0.000000 |
| egg-discrimination/6-attr_6-val_3-dist_0-seed | 110000 | 10000 | 11.000000 | 0.000000 |
| egg-discrimination/6-attr_6-val_3-dist_1-seed | 110000 | 10000 | 11.000000 | 0.000000 |
| egg-discrimination/6-attr_6-val_3-dist_2-seed | 110000 | 10000 | 11.000000 | 0.000000 |
| egg-discrimination/6-attr_6-val_9-dist_0-seed | 110000 | 10000 | 11.000000 | 0.000000 |
| egg-discrimination/6-attr_6-val_9-dist_1-seed | 110000 | 10000 | 11.000000 | 0.000000 |
| egg-discrimination/6-attr_6-val_9-dist_2-seed | 110000 | 10000 | 11.000000 | 0.000000 |
| egg-discrimination/8-attr_8-val_3-dist_0-seed | 110000 | 10000 | 11.000000 | 0.000000 |
| egg-discrimination/8-attr_8-val_3-dist_1-seed | 110000 | 10000 | 11.000000 | 0.000000 |
| egg-discrimination/8-attr_8-val_3-dist_2-seed | 110000 | 10000 | 11.000000 | 0.000000 |
| egg-reconstruction/4-attr_4-val_10-vocab_10-len | 110000 | 10000 | 11.000000 | 0.000000 |
| egg-reconstruction/6-attr_6-val_10-vocab_10-len | 110000 | 10000 | 11.000000 | 0.000000 |
| egg-reconstruction/8-attr_8-val_10-vocab_10-len | 110000 | 10000 | 11.000000 | 0.000000 |
| generalizations-mu-goodman/cub-concept | 1333330 | 133333 | 10.000000 | 0.000000 |
| generalizations-mu-goodman/cub-reference | 1333330 | 133333 | 10.000000 | 0.000000 |
| generalizations-mu-goodman/cub-set_reference | 1333330 | 133333 | 10.000000 | 0.000000 |
| generalizations-mu-goodman/shapeworld-concept | 1164800 | 166400 | 7.000000 | 0.000000 |
| generalizations-mu-goodman/shapeworld-reference | 1164800 | 166400 | 7.000000 | 0.000000 |
| generalizations-mu-goodman/shapeworld-set_reference | 1164800 | 166400 | 7.000000 | 0.000000 |
| nav-to-center/lexicon_size_11 | 65528 | 10000 | 6.552800 | 2.521193 |
| nav-to-center/lexicon_size_118 | 58664 | 10000 | 5.866400 | 2.167199 |
| nav-to-center/lexicon_size_17 | 59936 | 10000 | 5.993600 | 2.282533 |
| nav-to-center/lexicon_size_174 | 63129 | 10000 | 6.312900 | 2.434747 |
| nav-to-center/lexicon_size_25 | 61659 | 10000 | 6.165900 | 2.323010 |
| nav-to-center/lexicon_size_255 | 60054 | 10000 | 6.005400 | 2.263000 |
| nav-to-center/lexicon_size_37 | 62753 | 10000 | 6.275300 | 2.396604 |
| nav-to-center/lexicon_size_54 | 58778 | 10000 | 5.877800 | 2.197195 |
| nav-to-center/lexicon_size_7 | 61295 | 10000 | 6.129500 | 2.315368 |
| nav-to-center/lexicon_size_80 | 60250 | 10000 | 6.025000 | 2.256097 |
| nav-to-center/temperature_0.1 | 74939 | 10000 | 7.493900 | 3.180623 |
| nav-to-center/temperature_0.167 | 72255 | 10000 | 7.225500 | 2.995171 |
| nav-to-center/temperature_0.278 | 75732 | 10000 | 7.573200 | 3.106580 |
| nav-to-center/temperature_0.464 | 79810 | 10000 | 7.981000 | 3.564778 |
| nav-to-center/temperature_0.774 | 65665 | 10000 | 6.566500 | 2.527326 |
| nav-to-center/temperature_1.29 | 62566 | 10000 | 6.256600 | 2.364647 |
| nav-to-center/temperature_10 | 63105 | 10000 | 6.310500 | 2.397059 |
| nav-to-center/temperature_2.15 | 62019 | 10000 | 6.201900 | 2.314160 |
| nav-to-center/temperature_3.59 | 58786 | 10000 | 5.878600 | 2.187661 |
| nav-to-center/temperature_5.99 | 61106 | 10000 | 6.110600 | 2.289491 |
| rlupus/21-player.run-0 | 7131411 | 1001 | 7124.286713 | 445.837385 |
| rlupus/21-player.run-1 | 7196469 | 999 | 7203.672673 | 396.806665 |
| rlupus/21-player.run-2 | 7212723 | 1000 | 7212.723000 | 404.660045 |
| rlupus/9-player.run-0 | 565164 | 1003 | 563.473579 | 14.708875 |
| rlupus/9-player.run-1 | 417924 | 1010 | 413.786139 | 124.676603 |
| rlupus/9-player.run-2 | 414612 | 1000 | 414.612000 | 124.794461 |
| rlupus/9-player.run-3 | 416538 | 1003 | 415.292124 | 124.887337 |

Table 3

| name | Unique Tokens | Unique Lines | EoS Token Present | EoS Padding |
|---|---|---|---|---|
| babyai-sr/GoToObj | 5 | 653 | False | False |
| babyai-sr/GoToObjLocked | 6 | 788 | False | False |
| babyai-sr/GoToObjLocked_ambiguous | 6 | 1253 | False | False |
| babyai-sr/GoToObjLocked_ambiguous-freq_1 | 5 | 5171 | False | False |
| babyai-sr/GoToObjLocked_ambiguous-freq_2 | 4 | 3078 | False | False |
| babyai-sr/GoToObjLocked_ambiguous-freq_32 | 3 | 18 | False | False |
| babyai-sr/GoToObjLocked_ambiguous-freq_4 | 6 | 3241 | False | False |
| babyai-sr/GoToObjLocked_ambiguous-msg_16 | 9 | 4428 | False | False |
| babyai-sr/GoToObjLocked_ambiguous-msg_32 | 3 | 1887 | False | False |
| babyai-sr/GoToObjLocked_ambiguous-msg_4 | 2 | 1362 | False | False |
| babyai-sr/GoToObjUnlocked | 7 | 521 | False | False |
| babyai-sr/GoToObjUnlocked-freq_1 | 7 | 4614 | False | False |
| babyai-sr/GoToObjUnlocked-freq_2 | 8 | 3820 | False | False |
| babyai-sr/GoToObjUnlocked-freq_32 | 4 | 41 | False | False |
| babyai-sr/GoToObjUnlocked-freq_4 | 7 | 2766 | False | False |
| babyai-sr/GoToObjUnlocked-msg_16 | 13 | 1740 | False | False |
| babyai-sr/GoToObjUnlocked-msg_32 | 15 | 1430 | False | False |
| babyai-sr/GoToObjUnlocked-msg_4 | 3 | 400 | False | False |
| corpus-transfer-yao-et-al/cc | 391 | 309405 | True | True |
| corpus-transfer-yao-et-al/coco_2014 | 902 | 82783 | True | False |
| ec-at-scale/imagenet-10x1 | 20 | 161235 | False | False |
| ec-at-scale/imagenet-10x10 | 20 | 126775 | False | False |
| ec-at-scale/imagenet-1x1 | 20 | 145834 | False | False |
| ec-at-scale/imagenet-1x10 | 20 | 120182 | False | False |
| ec-at-scale/imagenet-5x5 | 20 | 169505 | False | False |
| egg-discrimination/4-attr_4-val_3-dist_0-seed | 10 | 240 | True | True |
| egg-discrimination/4-attr_4-val_3-dist_1-seed | 10 | 220 | True | True |
| egg-discrimination/4-attr_4-val_3-dist_2-seed | 9 | 187 | True | True |
| egg-discrimination/6-attr_6-val_3-dist_0-seed | 8 | 2326 | True | True |
| egg-discrimination/6-attr_6-val_3-dist_1-seed | 10 | 3279 | True | True |
| egg-discrimination/6-attr_6-val_3-dist_2-seed | 9 | 1976 | True | True |
| egg-discrimination/6-attr_6-val_9-dist_0-seed | 9 | 2883 | True | True |
| egg-discrimination/6-attr_6-val_9-dist_1-seed | 9 | 1015 | False | False |
| egg-discrimination/6-attr_6-val_9-dist_2-seed | 10 | 2499 | True | True |
| egg-discrimination/8-attr_8-val_3-dist_0-seed | 10 | 2610 | True | True |
| egg-discrimination/8-attr_8-val_3-dist_1-seed | 10 | 2789 | True | True |
| egg-discrimination/8-attr_8-val_3-dist_2-seed | 9 | 2656 | True | True |
| egg-reconstruction/4-attr_4-val_10-vocab_10-len | 7 | 228 | True | True |
| egg-reconstruction/6-attr_6-val_10-vocab_10-len | 8 | 1373 | True | True |
| egg-reconstruction/8-attr_8-val_10-vocab_10-len | 8 | 1464 | False | False |
| generalizations-mu-goodman/cub-concept | 23 | 27163 | False | False |
| generalizations-mu-goodman/cub-reference | 23 | 39457 | False | False |
| generalizations-mu-goodman/cub-set_reference | 23 | 35042 | False | False |
| generalizations-mu-goodman/shapeworld-concept | 17 | 12481 | False | False |
| generalizations-mu-goodman/shapeworld-reference | 17 | 7683 | False | False |
| generalizations-mu-goodman/shapeworld-set_reference | 17 | 28061 | True | False |
| nav-to-center/lexicon_size_11 | 8 | 2317 | False | False |
| nav-to-center/lexicon_size_118 | 61 | 4392 | False | False |
| nav-to-center/lexicon_size_17 | 15 | 3124 | False | False |
| nav-to-center/lexicon_size_174 | 40 | 3226 | False | False |
| nav-to-center/lexicon_size_25 | 12 | 1961 | False | False |
| nav-to-center/lexicon_size_255 | 37 | 3706 | False | False |
| nav-to-center/lexicon_size_37 | 22 | 2440 | False | False |
| nav-to-center/lexicon_size_54 | 43 | 4911 | False | False |
| nav-to-center/lexicon_size_7 | 7 | 1937 | False | False |
| nav-to-center/lexicon_size_80 | 35 | 3486 | False | False |
| nav-to-center/temperature_0.1 | 4 | 1437 | False | False |
| nav-to-center/temperature_0.167 | 4 | 1313 | False | False |
| nav-to-center/temperature_0.278 | 10 | 1308 | False | False |
| nav-to-center/temperature_0.464 | 4 | 1498 | False | False |
| nav-to-center/temperature_0.774 | 7 | 1639 | False | False |
| nav-to-center/temperature_1.29 | 9 | 2100 | False | False |
| nav-to-center/temperature_10 | 64 | 8793 | False | False |
| nav-to-center/temperature_2.15 | 64 | 8643 | False | False |
| nav-to-center/temperature_3.59 | 64 | 9044 | False | False |
| nav-to-center/temperature_5.99 | 64 | 9263 | False | False |
| rlupus/21-player.run-0 | 21 | 1001 | False | False |
| rlupus/21-player.run-1 | 21 | 999 | False | False |
| rlupus/21-player.run-2 | 21 | 1000 | False | False |
| rlupus/9-player.run-0 | 9 | 1003 | False | False |
| rlupus/9-player.run-1 | 9 | 1010 | False | False |
| rlupus/9-player.run-2 | 9 | 1000 | False | False |
| rlupus/9-player.run-3 | 9 | 1003 | False | False |

Table 4

| name | 1-gram Entropy | 1-gram Normalized Entropy | Entropy per Line |
| --- | --- | --- | --- |
| babyai-sr/GoToObj | 1.237631 | 0.533019 | 26.437867 |
| babyai-sr/GoToObjLocked | 0.986990 | 0.381820 | 47.817369 |
| babyai-sr/GoToObjLocked_ambiguous | 1.724020 | 0.666942 | 71.848479 |
| babyai-sr/GoToObjLocked_ambiguous-freq_1 | 1.463654 | 0.630362 | 736.239281 |
| babyai-sr/GoToObjLocked_ambiguous-freq_2 | 1.385921 | 0.692961 | 272.643237 |
| babyai-sr/GoToObjLocked_ambiguous-freq_32 | 0.358125 | 0.225952 | 4.381954 |
| babyai-sr/GoToObjLocked_ambiguous-freq_4 | 1.996955 | 0.772528 | 140.235868 |
| babyai-sr/GoToObjLocked_ambiguous-msg_16 | 2.555153 | 0.806061 | 237.183454 |
| babyai-sr/GoToObjLocked_ambiguous-msg_32 | 1.350560 | 0.852108 | 209.828108 |
| babyai-sr/GoToObjLocked_ambiguous-msg_4 | 0.922138 | 0.922138 | 16.612643 |
| babyai-sr/GoToObjUnlocked | 1.993155 | 0.709976 | 38.948688 |
| babyai-sr/GoToObjUnlocked-freq_1 | 0.896426 | 0.319313 | 248.726924 |
| babyai-sr/GoToObjUnlocked-freq_2 | 2.116083 | 0.705361 | 122.178237 |
| babyai-sr/GoToObjUnlocked-freq_32 | 1.643569 | 0.821785 | 13.315074 |
| babyai-sr/GoToObjUnlocked-freq_4 | 2.165184 | 0.771254 | 72.589650 |
| babyai-sr/GoToObjUnlocked-msg_16 | 2.608207 | 0.704837 | 117.930014 |
| babyai-sr/GoToObjUnlocked-msg_32 | 2.940985 | 0.752769 | 239.482412 |
| babyai-sr/GoToObjUnlocked-msg_4 | 1.453225 | 0.916883 | 14.784087 |
| corpus-transfer-yao-et-al/cc | 1.398306 | 0.162386 | 20.974592 |
| corpus-transfer-yao-et-al/coco_2014 | 6.599321 | 0.672235 | 98.989817 |
| ec-at-scale/imagenet-10x1 | 3.980879 | 0.921089 | 39.808790 |
| ec-at-scale/imagenet-10x10 | 3.908713 | 0.904391 | 39.087127 |
| ec-at-scale/imagenet-1x1 | 4.121796 | 0.953694 | 41.217964 |
| ec-at-scale/imagenet-1x10 | 3.975498 | 0.919844 | 39.754975 |
| ec-at-scale/imagenet-5x5 | 4.213196 | 0.974842 | 42.131963 |
| egg-discrimination/4-attr_4-val_3-dist_0-seed | 2.996740 | 0.902109 | 32.964144 |
| egg-discrimination/4-attr_4-val_3-dist_1-seed | 2.494699 | 0.750979 | 27.441685 |
| egg-discrimination/4-attr_4-val_3-dist_2-seed | 2.564778 | 0.809097 | 28.212561 |
| egg-discrimination/6-attr_6-val_3-dist_0-seed | 2.581470 | 0.860490 | 28.396171 |
| egg-discrimination/6-attr_6-val_3-dist_1-seed | 2.887394 | 0.869192 | 31.761330 |
| egg-discrimination/6-attr_6-val_3-dist_2-seed | 2.573849 | 0.811959 | 28.312341 |
| egg-discrimination/6-attr_6-val_9-dist_0-seed | 2.861929 | 0.902838 | 31.481224 |
| egg-discrimination/6-attr_6-val_9-dist_1-seed | 2.462500 | 0.776832 | 27.087504 |
| egg-discrimination/6-attr_6-val_9-dist_2-seed | 2.750845 | 0.828087 | 30.259294 |
| egg-discrimination/8-attr_8-val_3-dist_0-seed | 2.426752 | 0.730525 | 26.694277 |
| egg-discrimination/8-attr_8-val_3-dist_1-seed | 2.556315 | 0.769528 | 28.119469 |
| egg-discrimination/8-attr_8-val_3-dist_2-seed | 2.802140 | 0.883977 | 30.823535 |
| egg-reconstruction/4-attr_4-val_10-vocab_10-len | 2.296329 | 0.817969 | 25.259614 |
| egg-reconstruction/6-attr_6-val_10-vocab_10-len | 2.573243 | 0.857748 | 28.305674 |
| egg-reconstruction/8-attr_8-val_10-vocab_10-len | 2.295767 | 0.765256 | 25.253441 |
| generalizations-mu-goodman/cub-concept | 3.752944 | 0.829644 | 37.529443 |
| generalizations-mu-goodman/cub-reference | 3.103881 | 0.686159 | 31.038812 |
| generalizations-mu-goodman/cub-set_reference | 3.213538 | 0.710400 | 32.135376 |
| generalizations-mu-goodman/shapeworld-concept | 3.226724 | 0.789420 | 22.587066 |
| generalizations-mu-goodman/shapeworld-reference | 3.224439 | 0.788861 | 22.571074 |
| generalizations-mu-goodman/shapeworld-set_reference | 3.365556 | 0.823385 | 23.558893 |
| nav-to-center/lexicon_size_11 | 2.805418 | 0.935139 | 18.383341 |
| nav-to-center/lexicon_size_118 | 3.767532 | 0.635255 | 22.101847 |
| nav-to-center/lexicon_size_17 | 3.186153 | 0.815521 | 19.096524 |
| nav-to-center/lexicon_size_174 | 3.245330 | 0.609803 | 20.487443 |
| nav-to-center/lexicon_size_25 | 2.804201 | 0.782212 | 17.290421 |
| nav-to-center/lexicon_size_255 | 3.534679 | 0.678513 | 21.227163 |
| nav-to-center/lexicon_size_37 | 3.028477 | 0.679117 | 19.004602 |
| nav-to-center/lexicon_size_54 | 3.754792 | 0.691966 | 22.069917 |
| nav-to-center/lexicon_size_7 | 2.758577 | 0.982625 | 16.908697 |
| nav-to-center/lexicon_size_80 | 3.457586 | 0.674088 | 20.831957 |
| nav-to-center/temperature_0.1 | 1.994309 | 0.997155 | 14.945156 |
| nav-to-center/temperature_0.167 | 1.981753 | 0.990876 | 14.319155 |
| nav-to-center/temperature_0.278 | 1.986637 | 0.598037 | 15.045198 |
| nav-to-center/temperature_0.464 | 1.982692 | 0.991346 | 15.823868 |
| nav-to-center/temperature_0.774 | 2.311150 | 0.823248 | 15.176164 |
| nav-to-center/temperature_1.29 | 2.754878 | 0.869067 | 17.236170 |
| nav-to-center/temperature_10 | 4.905167 | 0.817528 | 30.954056 |
| nav-to-center/temperature_2.15 | 4.966695 | 0.827782 | 30.802945 |
| nav-to-center/temperature_3.59 | 5.340638 | 0.890106 | 31.395475 |
| nav-to-center/temperature_5.99 | 5.266347 | 0.877724 | 32.180539 |
| rlupus/21-player.run-0 | 4.062520 | 0.924915 | 28942.558214 |
| rlupus/21-player.run-1 | 4.196960 | 0.955523 | 30233.522552 |
| rlupus/21-player.run-2 | 3.997152 | 0.910033 | 28830.352466 |
| rlupus/9-player.run-0 | 3.079577 | 0.971498 | 1735.260160 |
| rlupus/9-player.run-1 | 3.119583 | 0.984119 | 1290.840311 |
| rlupus/9-player.run-2 | 3.090164 | 0.974838 | 1281.218984 |
| rlupus/9-player.run-3 | 3.111235 | 0.981485 | 1292.071343 |

Table 5

| name | 2-gram Entropy | 2-gram Conditional Entropy |
|---|---|---|
| babyai-sr/GoToObj | 1.544519 | 0.306888 |
| babyai-sr/GoToObjLocked | 1.147285 | 0.160295 |
| babyai-sr/GoToObjLocked_ambiguous | 1.978413 | 0.254393 |
| babyai-sr/GoToObjLocked_ambiguous-freq_1 | 2.071162 | 0.607508 |
| babyai-sr/GoToObjLocked_ambiguous-freq_2 | 1.538991 | 0.153070 |
| babyai-sr/GoToObjLocked_ambiguous-freq_32 | 0.420175 | 0.062050 |
| babyai-sr/GoToObjLocked_ambiguous-freq_4 | 2.406197 | 0.409242 |
| babyai-sr/GoToObjLocked_ambiguous-msg_16 | 3.097571 | 0.542418 |
| babyai-sr/GoToObjLocked_ambiguous-msg_32 | 1.463220 | 0.112660 |
| babyai-sr/GoToObjLocked_ambiguous-msg_4 | 1.717505 | 0.795367 |
| babyai-sr/GoToObjUnlocked | 2.497606 | 0.504450 |
| babyai-sr/GoToObjUnlocked-freq_1 | 1.092966 | 0.196541 |
| babyai-sr/GoToObjUnlocked-freq_2 | 2.877898 | 0.761815 |
| babyai-sr/GoToObjUnlocked-freq_32 | 1.731359 | 0.087790 |
| babyai-sr/GoToObjUnlocked-freq_4 | 2.979210 | 0.814026 |
| babyai-sr/GoToObjUnlocked-msg_16 | 3.043978 | 0.435771 |
| babyai-sr/GoToObjUnlocked-msg_32 | 3.157215 | 0.216230 |
| babyai-sr/GoToObjUnlocked-msg_4 | 2.307255 | 0.854029 |
| corpus-transfer-yao-et-al/cc | 2.059689 | 0.661383 |
| corpus-transfer-yao-et-al/coco_2014 | 12.884451 | 6.285130 |
| ec-at-scale/imagenet-10x1 | 6.811992 | 2.831113 |
| ec-at-scale/imagenet-10x10 | 6.328754 | 2.420041 |
| ec-at-scale/imagenet-1x1 | 6.882813 | 2.761016 |
| ec-at-scale/imagenet-1x10 | 6.375876 | 2.400379 |
| ec-at-scale/imagenet-5x5 | 7.137788 | 2.924591 |
| egg-discrimination/4-attr_4-val_3-dist_0-seed | 4.434835 | 1.438094 |
| egg-discrimination/4-attr_4-val_3-dist_1-seed | 3.550278 | 1.055580 |
| egg-discrimination/4-attr_4-val_3-dist_2-seed | 3.544613 | 0.979835 |
| egg-discrimination/6-attr_6-val_3-dist_0-seed | 3.917021 | 1.335551 |
| egg-discrimination/6-attr_6-val_3-dist_1-seed | 4.308021 | 1.420628 |
| egg-discrimination/6-attr_6-val_3-dist_2-seed | 3.738390 | 1.164541 |
| egg-discrimination/6-attr_6-val_9-dist_0-seed | 4.371053 | 1.509123 |
| egg-discrimination/6-attr_6-val_9-dist_1-seed | 3.578326 | 1.115826 |
| egg-discrimination/6-attr_6-val_9-dist_2-seed | 4.070906 | 1.320061 |
| egg-discrimination/8-attr_8-val_3-dist_0-seed | 3.504384 | 1.077631 |
| egg-discrimination/8-attr_8-val_3-dist_1-seed | 3.712531 | 1.156216 |
| egg-discrimination/8-attr_8-val_3-dist_2-seed | 4.006086 | 1.203946 |
| egg-reconstruction/4-attr_4-val_10-vocab_10-len | 3.212115 | 0.915787 |
| egg-reconstruction/6-attr_6-val_10-vocab_10-len | 3.750294 | 1.177051 |
| egg-reconstruction/8-attr_8-val_10-vocab_10-len | 3.515011 | 1.219244 |
| generalizations-mu-goodman/cub-concept | 5.686797 | 1.933852 |
| generalizations-mu-goodman/cub-reference | 5.641346 | 2.537465 |
| generalizations-mu-goodman/cub-set_reference | 5.509904 | 2.296366 |
| generalizations-mu-goodman/shapeworld-concept | 6.040857 | 2.814134 |
| generalizations-mu-goodman/shapeworld-reference | 5.908455 | 2.684016 |
| generalizations-mu-goodman/shapeworld-set_reference | 6.409305 | 3.043749 |
| nav-to-center/lexicon_size_11 | 4.240224 | 1.434806 |
| nav-to-center/lexicon_size_118 | 5.389004 | 1.621472 |
| nav-to-center/lexicon_size_17 | 4.655472 | 1.469319 |
| nav-to-center/lexicon_size_174 | 4.717891 | 1.472561 |
| nav-to-center/lexicon_size_25 | 4.106729 | 1.302528 |
| nav-to-center/lexicon_size_255 | 5.098629 | 1.563950 |
| nav-to-center/lexicon_size_37 | 4.335838 | 1.307361 |
| nav-to-center/lexicon_size_54 | 5.463441 | 1.708649 |
| nav-to-center/lexicon_size_7 | 4.123176 | 1.364599 |
| nav-to-center/lexicon_size_80 | 5.001934 | 1.544348 |
| nav-to-center/temperature_0.1 | 3.405157 | 1.410848 |
| nav-to-center/temperature_0.167 | 3.469046 | 1.487293 |
| nav-to-center/temperature_0.278 | 3.396763 | 1.410126 |
| nav-to-center/temperature_0.464 | 3.377160 | 1.394467 |
| nav-to-center/temperature_0.774 | 3.777791 | 1.466642 |
| nav-to-center/temperature_1.29 | 4.202502 | 1.447624 |
| nav-to-center/temperature_10 | 8.121348 | 3.216181 |
| nav-to-center/temperature_2.15 | 7.739814 | 2.773120 |
| nav-to-center/temperature_3.59 | 8.433494 | 3.092856 |
| nav-to-center/temperature_5.99 | 8.660965 | 3.394618 |
| rlupus/21-player.run-0 | 6.956412 | 2.893892 |
| rlupus/21-player.run-1 | 7.403071 | 3.206111 |
| rlupus/21-player.run-2 | 7.039882 | 3.042730 |
| rlupus/9-player.run-0 | 5.883233 | 2.803656 |
| rlupus/9-player.run-1 | 5.925070 | 2.805487 |
| rlupus/9-player.run-2 | 5.979073 | 2.888910 |
| rlupus/9-player.run-3 | 5.865222 | 2.753987 |

Table 6

