# OpenReview forum: "ELCC: the Emergent Language Corpus Collection"
_ICLR.cc/2025/Conference — Submitted to ICLR 2025_

### Official Review · Reviewer_HVVR · 2024-10-27

**Soundness:** 2
**Presentation:** 2
**Contribution:** 1
**Rating:** 1
**Confidence:** 3

**Summary:**

This paper is about providing a collection of artificial corpora resulting from "emergent communication systems". These systems generate artificial language by forms of interactions between agents (sender, receiver). Having such corpora is important for studying similarity of artificial language to human language - which in turn could be useful for obtaining pre-training data.

The main contribution of this the paper is the data resource and code for reproducibility.

**Strengths:**

This resource could be valuable for the niche community on artificial languages. The authors also identify reproducibility issues in related work.

**Weaknesses:**

While I do agree that these could be useful corpora and the paper should be published, I do not think that  ICLR is the right venue

- 1) it's mainly a resource contribution for which there are other venues, e.g., LREC or a corresponding journal

- 2) the whole field of study is quite niche

- 3) the substance is a little thin for an ICLR contribution, e.g., compiling approaches from other research

- 4) I would have hoped to read something in the context of recent LLM-based multi-agent interaction. However, mostly older approaches seemed to have been considered

- 5) There were some typos (minor), and I would also have liked to see a more central illustrating examples or an illustration figure show-casing agent-based simulation and language generation (e.g., the signalling game)

**Questions:**

As a suggestion, I would like to see an illustrating example. What are we supposed to be doing with the example in Figure 3? A sequence of integers is hard to read, interpret and understand.

What should we do with the numbers in Table 2?

Do I really understand correctly that you merely assemble the 73 corpora from other research and conduct a bit of analysis on it in Section 5?

---

> ### Author Response · Authors · 2024-11-16
>
> Thank you for your review for the paper and its assessment in the broader context
> of ICLR.
>
> > it's mainly a resource contribution for which there are other venues, e.g.,
> > LREC or a corresponding journal
>
> The [ICLR 2025 Call for Papers](https://iclr.cc/Conferences/2025/CallForPapers)
> lists "datasets and benchmarks" along with "reinforcement learning" under
> "relevant topics".  Additionally, ICLR has been one of the primary venues for
> publishing research on emergent communication more generally (e.g., [54 papers
> found for "emergent communication" in
> ICLR](https://openreview.net/search?term=%22emergent+communication%22&group=ICLR.cc&content=all&source=forum)).
>
> > Do I really understand correctly that you merely assemble the 73 corpora from
> > other research and conduct a bit of analysis on it in Section 5?
>
> The primary contribution of the paper is making a first-of-its-kind data
> resource available (emergent language corpora) available which incorporates the
> output of a wide variety of emergent communication systems.  We demonstrate
> some of the potential utility of this with a brief set of analyses.
>
>
> > the whole field of study is quite niche
>
> I would agree that emergent communication is a niche field, but that does not
> detract from its potential applications to NLP, linguistics, and beyond (e.g.,
> see Boldt & Mortensen ([TMLR
> 2024](https://openreview.net/forum?id=jesKcQxQ7j))).  As mentioned above, ICLR
> has been one of the primary venues for publishing research on deep
> learning-based emergent communication.
>
>
> > I would have hoped to read something in the context of recent LLM-based
> > multi-agent interaction. However, mostly older approaches seemed to have been
> > considered
>
> Much of emergent communication research is orthogonal to LLM-based approaches
> since we are trying observe language emerging _from scratch_, that is, without
> any kind of pretraining.  The majority of papers used for ELCC are from 2022 or
> 2023 and present ECSs that relevant for contemporary EC research (more so than
> LLM-based approaches to different problems).
>
>
> > As a suggestion, I would like to see an illustrating example. What are we
> > supposed to be doing with the example in Figure 3? A sequence of integers is
> > hard to read, interpret and understand.
>
> Figure 3 is primarily a qualitative example of (1) the format of emergent
> language corpora in ELCC, and (2) an example of what a high-performing vs.
> low-performing (on XferBench) language look like.  EC is difficult to interpret
> since there its meaning is invented through the RL process; interpreting such
> communication is an open problem in EC research.  To partially address this
> problem, we use tools like XferBench to assess characteristics like suitability
> for transfer learning to human languages.
>
>
> > There were some typos (minor)
>
> These will be fixed.
>
> > and I would also have liked to see a more central illustrating examples or an
> > illustration figure show-casing agent-based simulation and language
> > generation (e.g., the signalling game)
>
> We could include a figure of the signalling game.
>
>
> > What should we do with the numbers in Table 2?
>
> The primary purpose is to give a rough sense of the size and variety of corpora
> included in ELCC.  Interpreting emergent languages is an ongoing field of
> study, so we are limited looking at basic characteristics like entropy, token
> count, unique tokens, etc.

---

> ### Comment · Reviewer_HVVR · 2024-11-22
>
> Thanks a lot, dear authors!
>
> > (e.g., 54 papers found for "emergent communication" in ICLR).
>
> Do you have an idea what fraction of those papers has been accepted at the main conference? I see several withdrawn and submitted papers and papers going to workshops.
>
> I see this one for example has been accepted: https://openreview.net/pdf?id=rJxGLlBtwH
> but they do not merely assemble emergent language corpora.
>
> In general, I thank you for your answers but I feel this paper should be published somewhere else.
> I am not an expert for emergent communication, though, and if the other reviewers would strongly believe that this paper is useful and relevant for ICLR - merely as a collection of language corpora with some illustration of utility - I would say that my vote should be downweighed.
>
> My personal stance, however, is that this should not be sufficient for ICLR and that ICLR is still not the best (or even adequate) venue for this kind of research. There are tens of thousands of papers in ICLR (over the last several years) and it seems that a handful of emergent communication papers are among them, but typically I would expect to see such kind of research elsewhere.
>
> > Much of emergent communication research is orthogonal to LLM-based approaches since we are trying observe language emerging from scratch, that is, without any kind of pretraining.
>
> That's a pity. It would be great if some ingenious people from that field could find a way to bridge the communities though. I think I might be interested in such a paper.

---

### Official Review · Reviewer_98Vb · 2024-10-28

**Soundness:** 2
**Presentation:** 1
**Contribution:** 1
**Rating:** 3
**Confidence:** 4

**Summary:**

This paper presents a system and approach called Emergent Language Corpus Collection (ELCC) for the standardized storage of languages produced by emergent communication systems. It further discusses several commonly used coordination games, comparing the languages they generate primarily using the XferBench evaluation metric.

**Strengths:**

There is a clear need to standardize various aspects of emergent communication (EC) experimental setups, and efforts in this direction such as the suggested ELCC, are valuable. The authors provide a compelling critique of the difficulties in reproducing past EC research and identify missing elements that hinder accelerated progress. They call for a more robust framework to facilitate easier reproducibility, comparison, and competition in the field, which could significantly accelerate its development.

**Weaknesses:**

The motivation behind this work is underdeveloped. Although standardization is beneficial, it is unclear what specific problems this work aims to solve. A clear example of ELCC’s utility, such as its potential to generate new insights, would enhance its justification. Additionally, the static nature of the ELCC collection raises questions about the kinds of research it enables and what insights it may provide beyond those already published by the corresponding papers. A leaderboard with well-defined datasets, metrics, and results might be more impactful than a static collection of generated languages.

Furthermore, while the ELCC corpus could have been used to evaluate the soundness and completeness of various EC evaluation metrics proposed in the literature, the analysis is mostly limited to XferBench. For example, although the authors mention the HAS metric (Lines 464-468), they only discuss it theoretically rather than demonstrating how ELCC could aid in evaluating this metric across languages.

In conclusion, the paper lacks innovative contributions, and the provided resource appears too limited to address key challenges in the field, such as establishing standardized EC benchmarks with readily available datasets and evaluation metrics.

**Questions:**

Could the Hugging-Face platform, API, and cloud infrastructure be leveraged to host and promote ELCC to the broader research community?

---

> ### Author Response · Authors · 2024-11-16
>
> Thank you for your review and the suggested areas of improvement.
>
>
> > This paper presents a system and approach ... for the standardized storage of
> > languages produced by emergent communication systems.
>
> Just to add to the summary given, the primary contribution of this paper is
> providing a well-organized resource: that is collection of EL corpora generated
> from across the EC literature.
>
>
> > They call for a more robust framework to facilitate easier reproducibility,
> > comparison, and competition in the field, which could significantly
> > accelerate its development.
>
> We do agree with this statement but would like to add to that we argue not just
> for the necessity of a framework but for an actual resource that makes EL data
> publicly available, such as ELCC.
>
>
> > A clear example of ELCC’s utility, such as its potential to generate new
> > insights, would enhance its justification.
>
> Although we do not focus on generating new insights directly since this is
> a dataset paper, we give examples of insights in Section 5 that would not be
> possible without a collection of EL corpora like ELCC.  One example to
> highlight from the section would be Figure 4a which shows that there
> a consistent correlation between the entropy of an emergent language corpus and
> its suitability for transfer learning to human language.  ELCC makes it
> possible to observe this trend across a variety of emergent language
> environments and implementations, suggesting that it might be a general
> property and not the result of an implementation detail.
>
>
> > Additionally, the static nature of the ELCC collection raises questions about
> > the kinds of research it enables and what insights it may provide beyond
> > those already published by the corresponding papers.
>
> We do agree that the lack of semantic annotations is limitation of ELCC,
> although not one that precludes all potential applications.
>
>
> > A leaderboard with well-defined datasets, metrics, and results might be more
> > impactful than a static collection of generated languages.
>
> ELCC would be a key component in generating baselines for a leaderboard based
> on a metric like XferBench.  Figure 2 presents such results (although not in
> a leaderboard arrangement).
>
>
> > Furthermore, while the ELCC corpus could have been used to evaluate the
> > soundness and completeness of various EC evaluation metrics proposed in the
> > literature, the analysis is mostly limited to XferBench. For example,
> > although the authors mention the HAS metric (Lines 464-468), they only
> > discuss it theoretically rather than demonstrating how ELCC could aid in
> > evaluating this metric across languages.
>
> We do agree that the analyses presented in this paper are limited.  XferBench
> was the primary metric used for analysis because it offers a ready-to-run open
> source implementation that already accepts the data format used by ELCC.
>
>
> > In conclusion, the paper lacks innovative contributions, and the provided
> > resource appears too limited to address key challenges in the field, such as
> > establishing standardized EC benchmarks with readily available datasets and
> > evaluation metrics.
>
> While we agree with the fact that ELCC cannot address all of the key challenges
> in the field of EC, we would argue that ELCC is an important component (and one
> of the last missing pieces) in establishing a benchmark (while the standardized
> evaluation data and metrics are complemented by approaches such as XferBench).
>
>
> > Could the Hugging-Face platform, API, and cloud infrastructure be leveraged
> > to host and promote ELCC to the broader research community?
>
> Yes, we intend to leverage a service like HuggingFace Datasets for the
> distribution of ELCC, although we opted not to use it for review due to
> anonymity constraints.

---

> > ### Comment · Reviewer_98Vb · 2024-11-18
> > **Reply to Authors**
> >
> > I appreciate the authors' responses to my questions.
> >
> > While I believe the current version of the paper is not yet ready for publication, I encourage the authors to build upon their work with ELCC to create a comprehensive set of benchmarks for the EC community. Such benchmarks, ideally comprising multiple datasets and standardized evaluation metrics, could provide significant value to the field. Furthermore, making the ELCC resources available for download through Hugging Face would greatly enhance accessibility for the community.
> >
> > Based on the provided answers, I have decided to maintain my score.

---

> > > ### Author Response · Authors · 2024-11-18
> > > **Clarification question**
> > >
> > > Would you be able to clarify what a "comprehensive set of benchmarks" means in this case as well as "multiple datasets and standardized evaluation metrics"?  ELCC itself is not a benchmark per se but a set of datasets (but maybe different from the "multiple datasets" mentioned in the review?), and we do test it on the only standardized evaluation metric available in the field of EC (i.e., XferBench).  This would be helpful in understanding what to improve in the paper.  Thanks.

---

> > > > ### Comment · Reviewer_98Vb · 2024-11-18
> > > > **Second reply to authors**
> > > >
> > > > By "comprehensive set of benchmarks," I refer to datasets that have been previously utilized by the emergent communication (EC) community for conducting experiments. For instance, two examples can be found in J. Mu and N. Goodman's *Emergent Communication of Generalizations* (NeurIPS, 2021). However, numerous other examples exist throughout the literature.
> > > >
> > > > By "standardized evaluation metrics available in the field," I mean established measures such as:
> > > >
> > > > - **Topographic Similarity**, widely used in the field.
> > > > - **PosDis and BosDis** metrics proposed by Chaabouni et al. in *Compositionality and Generalization in Emergent Languages*.
> > > > - **CI (Compositionality Index)** introduced by Bogin et al. in *Emergence of Communication in an Interactive World with Consistent Speakers*.
> > > > - **CBM (Concept-Best-Matching)** developed by Carmeli et al. in *Concept-Best-Matching: Evaluating Compositionality in Emergent Communication*.
> > > >
> > > > And other similar metrics found in the literature.

---

### Official Review · Reviewer_E7aW · 2024-11-04

**Soundness:** 4
**Presentation:** 4
**Contribution:** 3
**Rating:** 6
**Confidence:** 3

**Summary:**

This paper introduces a collection of "corpora" called Emergent Language Corpus Collection (ELCC), generated from open-source emergent communication systems (ECS). In Emergent Communication (EC), researchers are generally required to be familiar with deep learning, reinforcement learning, and machine learning frameworks such as PyTorch to simulate the emergence of language and communication. However, not all of the previous EC studies provide well-documented, high-quality, and open-source implementations, which prevents us from performing the meta-analysis of the EC studies and non-ML researchers from joining the EC field. To mitigate such problems, this paper introduces ELCC, collecting several appropriate open-source implementations, conducting reproduction experiments, summarizing the experimental results (i.e., emergent languages) as "corpora," and performing some basic evaluations (e.g., n-gram entropy, XferBench). The author(s) suggest that ELCC can also be used for further analyses (e.g., word boundaries, syntax).

**Strengths:**

- Contribution to the reproducibility of the EC field and the ease of entry to the field.
- The novelty of the idea of creating corpora of emergent languages, which had not been conceptualized before, probably due to the artificial nature of emergent languages.
- Most EC papers mention previous work in the "Background" section or "Related Work" section regarding problem setting, methodology, and evaluation metrics. Interestingly, this paper additionally focuses on the quality and availability of their source code implementations and summarizes them comprehensively, which will be helpful for EC researchers.
- The corpus collection process is carefully planned and conducted.
- The paper is clearly written and easy to read.

**Weaknesses:**

- ELCC is not annotated with meanings (or semantics, inputs) corresponding to messages (sentences). This would be somewhat problematic when the EC researchers want to evaluate other properties, such as compositionality.
- The basic analyses provided (e.g., XferBench) are indeed valid in showing the "negative result" (i.e., that emergent language is not similar to human language). However, I am not sure if the provided metrics are still valid in showing the "positive result" in the future (i.e., that emergent language is similar enough to human language), recalling some (computational) linguistic literature claiming that even monkey-typing sequences follow a Zipf-like distribution [1], that even pre-training with simple artificial language ($\neq$ emergent language) has a positive influence on downstream task performance [2], etc. This is another reason I think ELCC should be annotated with semantic information to allow future EC researchers to perform more detailed analyses.
- I feel references (only) to Ueda et al. (2023) and van der Wal et al. (2019) in some parts of the Introduction and Discussion sections are somewhat abrupt, considering a number of  EC papers have studied human language-ness of emergent languages from various points of view. I know typical topics in the field, like TopSim-based compositionality, are out of this paper's scope. Still, I'd be glad if you mention a few more human-language-ness papers somewhere in this paper, e.g., in the Limitation section.
- (minor) Figure 4 has no caption (other than sub-captions a, b).
- (minor) When mentioning Zipf's law, you may cite Zipf (1949) [3]. Likewise, you may cite Harris (1955) [4] and Tanaka-Ishii (2021) [5] for Harris' articulation scheme.

[1] Piantadosi, S.T. Zipf's word frequency law in natural language: A critical review and future directions. Psychon Bull Rev 21, 1112–1130 (2014). https://doi.org/10.3758/s13423-014-0585-6

[2] Ryokan Ri and Yoshimasa Tsuruoka. 2022. Pretraining with Artificial Language: Studying Transferable Knowledge in Language Models. In Proceedings of the 60th Annual Meeting of the Association for Computational Linguistics (Volume 1: Long Papers), pages 7302–7315, Dublin, Ireland. Association for Computational Linguistics.

[3] Zipf, G. K. (1949). Human behavior and the principle of least effort. Addison-Wesley Press.

[4] Zellig S. Harris. From phoneme to morpheme. Language, 31(2):190–222, 1955. URL http://www.jstor.org/stable/411036.

[5] Kumiko Tanaka-Ishii. Articulation of Elements, pp. 115–124. Springer International Publishing,
Cham, 2021. URL https://doi.org/10.1007/978-3-030-59377-3_11.

**Questions:**

- Is it possible to (easily) update ELCC with semantic annotations in the (near) future?
- With ELCC, is it possible to evaluate language similarity (or language synchronization), which is sometimes considered in the context of populated signaling games?
- What can we read from Table 2? Does it indicate either language-ness or unlikeness of emergent language?
- (minor; just curious) As EC researchers, how can we contribute to this corpora creation project after deanonymization (e.g., by sending a pull request)?

---

> ### Author Response · Authors · 2024-11-16
>
> Thank you for your thorough review.  We largely agree with above assessment of
> the strengths and weaknesses of the paper.
>
> > However, I am not sure if the provided metrics are still valid in showing
> > the "positive result" in the future (i.e., that emergent language is similar
> > enough to human language), ...
>
> We agree that this paper does not present any evidence sufficient to make the
> claim that emergent language will catch up to human language at some point.  We
> do not attempt to make this claim on empirical grounds although some of the
> overall motivation comes from the intuitive viability of the field.  If there
> is a specific place that you think we overstate our case, we can remediate that
> portion.
>
>
> > I feel references (only) to Ueda et al. (2023) and van der Wal et al. (2019)
> > in some parts of the Introduction and Discussion sections are somewhat
> > abrupt, considering a number of EC papers have studied human language-ness of
> > emergent languages from various points of view. I know typical topics in the
> > field, like TopSim-based compositionality, are out of this paper's scope.
> > Still, I'd be glad if you mention a few more human-language-ness papers
> > somewhere in this paper, e.g., in the Limitation section.
>
> Noted.  We highlight those approaches in particular because although they are
> not representative of the majority of EC research, they are the approaches that
> could most readily benefit from a resource like ELCC.  As suggested, we can
> discuss how more representative approaches (e.g., toposim, compositionality
> generally), do or do not fit in with what ELCC provides.
>
>
> > (minor) When mentioning Zipf's law, you may cite Zipf (1949) [3]. Likewise,
> > you may cite Harris (1955) [4] and Tanaka-Ishii (2021) [5] for Harris'
> > articulation scheme.
>
> Thank you.  We will add these references.
>
>
> > Is it possible to (easily) update ELCC with semantic annotations in the
> > (near) future?
>
> Yes, I believe it is possible as much of the effort of this paper went into
> collecting and reproducing the codebases used to generate the corpora.  Given
> that ELCC has already curated and improved the reproducibility of these
> codebases, collecting the semantic annotations should require less of this
> "dirty work".  The main challenge in producing semantic annotations, then, is
> defining a formalism that can both capture the flexibility of different EC
> systems while at the same time providing enough consistency to make
> general-purpose analyses feasible and meaningful.
>
>
> > With ELCC, is it possible to evaluate language similarity (or language
> > synchronization), which is sometimes considered in the context of populated
> > signaling games?
>
> There is certainly some types of similarity that can be assessed with ELCC.
> These types would focus on statistical and distributional similarity, including
> EL-to-EL transfer learning suitability with something like XferBench.  Some
> more exotic approaches to this could include training token embeddings on the
> EL corpora with methods like word2vec and comparing the similarity of the
> embedding spaces.
>
> I am not quite sure what "language synchronization" refers to, so if there is
> a particular formulation, a pointer towards that could help me address the
> question better.  As stated in the review though, I think the gold standard of
> measuring language similarity will require semantic annotations, going beyond
> purely distributional properties.
>
>
> > What can we read from Table 2? Does it indicate either language-ness or
> > unlikeness of emergent language?
>
> The main purpose of this table is to give a sense of the "shape" of emergent
> languages in the corpus, so that readers can have a sense of how big or how
> small the corpora all; directly comparing these values to human language was
> not the original intent.  That being said, it might be helpful to include
> figures from a human language dataset or two just to give a point of comparison
> (e.g., a human language Wikipedia-derived dataset might have an entropy of
> around 12 bits after being tokenized with BPE compared to the max emergent
> language value of 6.6).
>
>
> > (minor; just curious) As EC researchers, how can we contribute to this
> > corpora creation project after deanonymization (e.g., by sending a pull
> > request)?
>
> Thank you for asking.  We intend to publicize ELCC with a platform like
> HuggingFace Datasets or some equivalent that will permit not only easy access
> but easy contributions, as we will welcome such engagement with ELCC.

---

> > ### Comment · Reviewer_E7aW · 2024-11-21
> > **Reply to authors**
> >
> > Thank you for providing detailed responses to my comments.
> >
> > > I am not quite sure what "language synchronization" refers to, so if there is a particular formulation, a pointer towards that could help me address the question better.
> >
> > I meant the measure used in [6, 7].
> > It measures to what extent messages from the same inputs (but generated by different sender agents) are similar (w.r.t edit distance).
> >
> > [6] Mathieu Rita, Florian Strub, Jean-Bastien Grill, Olivier Pietquin, & Emmanuel Dupoux (2022). On the role of population heterogeneity in emergent communication. In International Conference on Learning Representations.
> >
> > [7] Paul Michel, Mathieu Rita, Kory Wallace Mathewson, Olivier Tieleman, & Angeliki Lazaridou (2023). Revisiting Populations in Multi-agent Communication. In International Conference on Learning Representations.
> >
> > I maintain the current score, as I have already taken a positive stance on this paper.

---

### Official Review · Reviewer_CVc2 · 2024-11-04

**Soundness:** 4
**Presentation:** 4
**Contribution:** 3
**Rating:** 6
**Confidence:** 3

**Summary:**

The paper introduces the Emergent Language Corpus Collection (ELCC), a collection of corpora generated from open-source implementations of emergent communication systems (ECS). The authors perform quantitative and qualitative analyses with ELCC to demonstrate potential use cases of the resource.

**Strengths:**

1. The sources and statistics of the data are well documented.
2. The data can enable broader engagement and new research directions in ECS.

**Weaknesses:**

1. As the authors already mentioned in limitations, the data are not annotated, having not reference to the semantics of the communication. This limits the scope of possible analysis.

**Questions:**

1. "it is not comprehensive collection of all of the open source implementations let alone all ECSs in the literature" -- to your best knowledge, what are some ECS categories having no representative systems in ELCC?

2. Typo: line 169: syntehtic -> synthetic

---

> ### Author Response · Authors · 2024-11-16
>
> Thank you for the review of this paper.
>
> > "it is not comprehensive collection of all of the open source implementations
> > let alone all ECSs in the literature" -- to your best knowledge, what are
> > some ECS categories having no representative systems in ELCC?
>
> Two of the most significant gaps would be embodied, multi-agent, multi-step
> games that go beyond simple navigation such as Mordatch & Abeel (2017,
> https://arxiv.org/abs/1703.04908) as well as negotiation-based conversational
> games such as Cao & al. (2018, https://openreview.net/forum?id=Hk6WhagRW).
> Open source implementations of these games are not available, making their
> inclusion in ELCC difficult.  We can highlight these in the paper.
>
>
> > Typo: line 169: syntehtic -> synthetic
>
> We will fix this.

---

### Meta-Review · Area_Chair_7gev · 2024-12-20

**Metareview:**

This submission presents an interesting contribution to the community, and the proposed corpora have potential utility. However, several concerns raised by the reviewers remain unresolved:

- The paper lacks significant innovative contributions.
- The benchmark provided does not adequately address key challenges in the field, such as the need for standardized EC benchmarks with readily accessible datasets and evaluation metrics.
- The substance of the work appears limited; the benchmark is assembled from existing datasets without introducing new approaches.
- It is unclear which specific research directions the benchmark facilitates and what novel insights it offers beyond those already discussed in the referenced papers.

While we appreciate the authors' efforts to address these concerns during the rebuttal process, the reviewers agree that the contributions of this submission fall short of the standard expected at ICLR.

**Additional Comments On Reviewer Discussion:**

No changes. The meta-review summarized unsolved points raised by the reviewers.

---

### Decision · Program_Chairs · 2025-01-22

Reject